# Gingipain Genotyping as a Potential Predictor for the Assessment of Periodontal Health and Disease Condition

Manohar Kugaji [1] , Kishore Bhat [1], Uday Muddapur [2] , Vinayak Joshi [3,*], Malleswara Rao Peram [4] and Vijay Kumbar [1]

1   Central Research Laboratory, Maratha Mandal's Nathajirao G. Halgekar Institute of Dental Sciences & Research Centre, Belagavi 590010, India
2   Department of Biotechnology, KLE Technological University, Vidyanagar, Hubballi 5800031, India
3   Department of Periodontics, School of Dentistry, Louisiana State University, New Orleans, LA 70119, USA
4   Department of Pharmaceutics, Chebrolu Hanumaiah Institute of Pharmaceutical Sciences, Guntur 522019, India
*   Correspondence: vjosh1@lsuhsc.edu

**Abstract:** Oral hygiene maintenance is important to maintain optimal oral health. Oral health is affected by dysbiotic oral microflora in the dental plaque. Virulent factors of pathogenic organisms, such as gingipain, are responsible for tissue degradation and host tissue invasion in periodontal disease. We sought to investigate the distribution of gingipain genotypes (*rgpA* and *kgp*) of *P. gingivalis* in patients with chronic periodontitis and healthy individuals. The study included individuals positive for *P. gingivalis*, with 95 samples in the chronic periodontitis (CP) group and 35 samples in the healthy (H) group. We found that *kgp-I* and *kgp-II* types were prevalent in 67.36% and 32.64% of the samples in the CP group, respectively. In the H group, *kgp-II* was highly prevalent (97.14%). The *rgpA* genotype, type A was found in 78.95% and 82.85% of the samples in the CP and H group, respectively. The mean level of PD and CAL were increased in the presence of *kgp-I* and decreased in the presence of *kgp-II*. The mean level of *P. gingivalis* was increased in the presence of *kgp-I* and *rgpA*, type A. Our results show that *kgp-I* and *kgp-II* are strongly associated with disease and health condition, respectively.

**Keywords:** cysteine protease; chronic periodontitis; genotyping; oral health; PCR; RFLP

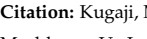



## 1. Introduction

Periodontitis is a polymicrobial infectious disease characterized by inflammation within supporting tissues of teeth, loss of clinical attachment and alveolar bone [1]. Putative pathogens in the oral biofilm interact with host tissues and contribute to an exaggerated inflammatory response, leading to periodontitis [2]. *Porphyromonas gingivalis* defined as a 'keystone pathogen' for its ability to disrupt the host protective mechanism is considered a putative causative agent responsible for the pathogenesis and development of chronic periodontitis [3]. The pathogenic capability of *P. gingivalis* is reflected in its diverse virulent factors, such as fimbriae, cysteine proteases (gingipains), capsules, haemagglutinins, lipopolysaccharides, exopolysaccharides, outer membrane proteins, outer membrane vesicles and lipoteichoic acids. These virulent factors aid the organism in colonization and to evade antibacterial host defense mechanisms, and damage host periodontal structure [4,5]. Gingipains are one of the important pathogenic agents of *P. gingivalis* and are being studied extensively for their key role in contributing to pathogenesis.

Gingipains are trypsin-like cysteine proteinases belonging to the peptidase family C25 which are broadly categorized as arginine-dependent gingipain R (protein Rgp) and lysine-dependent gingipain K (protein Kgp), based on their ability to incise polypeptides after arginine and lysine residues, respectively [6]. The gene coding for gingipain R containing hemagglutinin/adhesion (HA) domain is called *rgpA/prpR1*, whereas the gene that encodes

gingipain R without an HA domain is referred to as *rgpB*. Allaker et al. identified three genotypes of *rgpA/prpR1* gingipain (type A, B, and C) [7]. *kgp* is referred to as the gene encoding gingipain. K. Beikler et al. reported two diverse types of *kgp* (*kgp-I* and *kgp-II*) according to the sequence dissimilarity in the region that encodes the catalytic domain [8].

Gingipains are involved in haem acquisition which is required for bacterial growth. *P. gingivalis* acquires haem mainly from haemoglobin. The HA region serves a vital role in the haemoglobin-binding strength of RgpA and Kgp [9], while RgpB lacking in the HA region has minimal binding activity [10]. Gingipains, are similar to extracellular matrix proteins, plasma proteins, cytokines, and host cell surface proteins in their ability to cleave and degrade host proteins [11]. Gingipains contribute to bleeding owing to the role of Kgp in the efficient digestion of fibrinogen and fibrin, which are responsible for blood coagulation [12]. It has also been shown that Rgp upregulates MMP-1 expression from gingival fibroblasts and directly activates MMPs, which contributes to destruction of the matrix tissue [13].

It has been observed that virulence can differ at both the level of the genotype and in the phenotypic expression of the gene [14,15]. The difference in pathogenic ability among the strains of *P. gingivalis* observed are attributed to the presence of specific virulence genes. The disparity in the virulent activity results in a differential level of severity in oral health and disease. Thus, across the globe, study of virulence factors has become an important issue for understanding disease etiology.

The distribution and role of gingipain genotypes is well established and studied in diseased conditions; however, only a few reports on the distribution of genotypes in periodontal health exist. Hence, we sought to investigate the occurrence of gingipain genotypes of *P. gingivalis* in patients with chronic periodontitis and healthy individuals and to assess their association with clinical characteristics in these subjects.

## 2. Materials and Methods

### 2.1. Sample Collection

The study included 95 samples from the chronic periodontitis (CP) group and 35 samples from the healthy (H) group, all of which were positive for *P. gingivalis* in our earlier reported analysis of 120 samples from each group [16]. All the participants included in the study were screened to meet the inclusion and exclusion criteria as reported in our earlier paper. Probing depth (PD), clinical attachment loss (CAL), probing index (PI) and bleeding index (BI) were recorded for each participant. Subgingival plaque samples were collected using a universal curette from all the teeth except the third molars. DNA extraction and quantification of *P. gingivalis* by real-time PCR was performed as described in [16].

### 2.2. Amplification of kgp and rgpA Gene

The amplification of *kgp* and *rgpA* genes was performed as described by Beikler et al. and Allaker et al., respectively [7,8]. The primer pairs with their amplified length were as described in Table 1. A reaction mixture was prepared in 0.2 mL PCR tubes with a total volume of 25 μL for each sample. To the PCR tube, 12.5 μL of Ampliqon red, 2X master mix (Ampliqon, Odense, Denmark) was added containing Tris-HCL pH 8.5, $(NH_4)_2SO_4$, 3 mM $MgCl_2$, 0.2% Tween 20, 0.4 mM of each dNTP, 0.2 units/μL Ampliqon Taq DNA polymerase, and inert red dye and stabilizer. Then 2μL of each of the primers specific to *kgp* and *rgpA* were added to the separate reactions from a working concentration of 20 pmole/μL. The DNA template was added at approximately 100 ng concentration.

For amplification of the *kgp* gene, the thermal cycling conditions were as follows: The initial denaturation was performed at 95 °C for 3 min, followed by 45 cycles of denaturation at 95 °C for 1 min, annealing at 62 °C for 1 min and extension at 72 °C for 2 min. Final extension was carried out at 72 °C for 5 min. For amplification of the *rgpA* gene, the thermal cycling conditions used were the same as used for the *kgp* gene except that the extension step was carried out at 72 °C for 3 min.

**Table 1.** PCR primers used for gingipain genotyping (*kgp* and *rgpA*) by PCR.

| Gingipains | Primer Sequences (5′-3′) | Product Length in Base Pair | Reference |
|---|---|---|---|
| *kgp* forward | GAACTGACGAACATCATTG | 890 | [8] |
| *kgp* reverse | GCTGGCATTAGCAACAC | | |
| *rgpA* forward | AGTGAGCGAAACTTCGGA | 1700 | [7] |
| *rgpA* reverse | GGTATCACTGGGTATAACCTGT | | |

PCR product length of 890 base pairs specific for the *kgp* gene and 1700 base pairs specific for the *rgpA* gene were identified by subjecting PCR amplified samples to 2% agarose gel electrophoresis, followed by staining with ethidium bromide (0.5 μg/mL). The remaining aliquot of PCR product was further used for *kgp* and *rgpA* genotyping by restriction fragment length polymorphism (RFLP).

*2.3. Restriction Digestion for kgp and rgpA Genotyping*

The *kgp* gene product was digested with the restriction enzyme Fastdigest MseI (Tru1I) (Thermo Scientific, Waltham, MA, USA). The reaction mixture was allowed to incubate at 65 °C for 5 min. The *rgpA* gene product was digested with the enzyme Fastdigest RsaI (Thermo Scientific, MA, USA). The reaction mixture was incubated at 37 °C for 5 min. The digested products were further separated on 3% agarose gel electrophoresis. The gel was stained with ethidium bromide and the bands were visualized on a UV gel documentation system (Major Science, Saratoga, NY, USA).

*2.4. Statistical Analysis*

Statistical analysis was carried out using GraphPad Prism 5.1 (GraphPad Software, Inc., San Diego, CA, USA). The frequency of the *kgp* and *rgpA* genotypes, and their statistical association with chronic periodontitis, was evaluated using Fisher's exact test. The relationship of the genotypes with clinical parameters and the quantity of *P. gingivalis* was examined using an unpaired *t*-test and the Mann Whitney U test, respectively. $p < 0.05$ was considered as statistically significant.

**3. Results**

In the present study, we processed 95 samples from the CP group and 35 samples from the H group, which were found to be positive for *P. gingivalis* by real-time PCR in our earlier study [16]. The *kgp* genotyping in the CP group showed that kgp type I was more prevalent (67.36%), followed by *kgp-II* (32.64%), whereas in the H group, *kgp-II* was more prevalent (97.14%) than *kgp-I* (2.8%). Hence, *kgp-I* was significantly associated with the CP group (*p* value < 0.0001), whereas there was a significant association of *kgp-II* with the H group (*p* value < 0.0001). In the *rgpA* genotyping in the CP group, type A was more prevalent (78.95%), followed by type B and type C (16.84% and 4.2%, respectively). In the H group similar results were obtained with type A and type B being present in 82.85% and 17.14% of the samples, respectively. Type C was not found in any of the samples. There was no statistical association found for *rgpA* genotypes with the CP or H group (Table 2) (Figure 1).

**Table 2.** Frequency distribution of gingipain genotypes in chronic periodontitis and healthy groups.

| Gingipain Genotypes | Chronic Periodontitis | | Healthy | | Odd Ratio | 95% Confidence Interval | *p* Value |
|---|---|---|---|---|---|---|---|
| | N | % | N | % | | | |
| *kgp-I* | 64 | 67.36 | 1 | 2.8 | 65.65 | 19.33 to 222.9 | <0.0001 * |
| *kgp-II* | 31 | 32.64 | 34 | 97.14 | 0.015 | 0.004 to 0.05 | <0.0001 * |
| *rgpA*, type A | 75 | 78.95 | 29 | 82.85 | 0.77 | 0.37 to 1.56 | 0.8055 |
| *rgpA*, type B | 16 | 16.84 | 6 | 17.14 | 1 | 0.47 to 2.09 | 1 |

* $p < 0.05$.

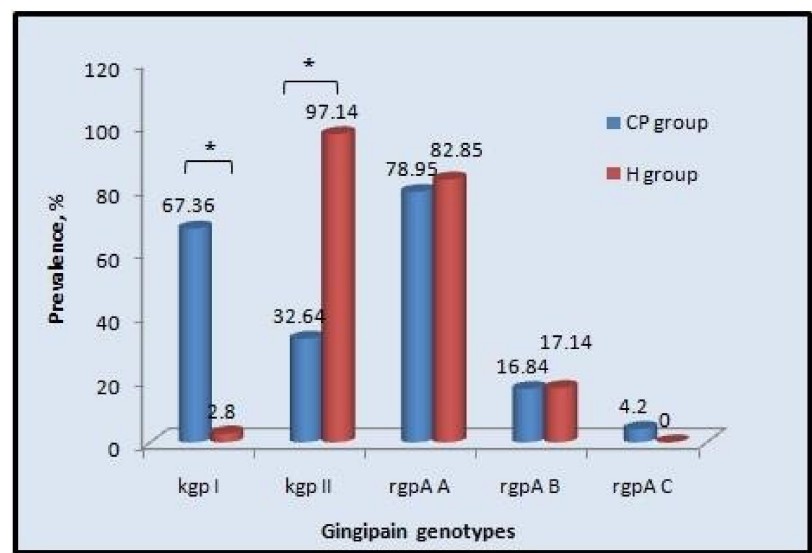

**Figure 1.** Barchart depicting distribution of gingipain genotypes in chronic periodontitis and healthy groups. * $p < 0.05$.

The presence of different genotypes of *kgp* and *rgpA* were correlated with clinical parameters. The mean level of PD in the presence of *kgp-I* was 5.88 ± 0.74 compared to its absence (5.45 ± 0.38). The mean level of CAL was 5.64 ± 1.27 in the presence of *kgp-I* when compared to the absence of this genotype (4.71 ± 1.20). There was a statistical association between the presence of *kgp-I* with PD and CAL with a *p* value < 0.0001 for both. The mean level of PD in the presence of *kgp-II* was 5.35 ± 0.52 when compared to the absence of *kgp-II* (5.8 ± 0.56). Similarly, the mean CAL level was 4.74 ± 1.49 in the presence of *kgp-II* when compared to its absence (5.37 ± 1.16). There was a significant difference for the absence of *kgp-II* with PD and CAL (*p* value < 0.0001 and 0.004, respectively). For the presence of *rgpA*, type A there was a higher mean level of CAL (5.35 ± 1.21) and PI (2.54 ± 0.21) when compared to its absence. There was a noteworthy association between the presence of *rgpA* type A with CAL and PI. In the presence of *rgpA* type C, we found the mean level of PI was 2.30 ± 0.40 when compared to the absence of *rgpA* type C (2.52 ± 0.18). The difference was statistically significant (*p* value 0.02) (Table 3).

When the quantitative data of *P. gingivalis* were correlated with the presence or absence of specific genotypes of *kgp* and *rgpA*, we found that the mean level of *P. gingivalis* in the presence of *kgp-I* was $2.11 \times 10^8$ (median $3.85 \times 10^6$) compared to $7.51 \times 10^7$ (median $1.89 \times 10^4$) in the absence of this genotype. Hence the quantity of *P. gingivalis* was significantly associated with the presence of *kgp* type I (*p* value < 0.0001). The mean level of *P. gingivalis* in the presence of *rgpA* type A was $1.57 \times 10^8$ (median $4.07 \times 10^6$) compared to $1.32 \times 10^8$ (median 0.0) in the absence of type A. The difference was statistically significant with a *p* value < 0.0001 (Table 4).

**Table 3.** Association of genotypes of gingipain with clinical parameters in chronic periodontitis patients.

| Gingipain Genotypes | Present/ Absent | PD | | CAL | | PI | | GI | |
|---|---|---|---|---|---|---|---|---|---|
| | | Mean ± SD | *p* Value | Mean ± SD | *p* Value | Mean ± SD | *p* Value | Mean ± SD | *p* Value |
| *kgp-I* | Present | 5.88 ± 0.74 | <0.0001 * | 5.64 ± 1.27 | <0.0001 * | 2.51 ± 0.23 | 0.865 | 2.55 ± 0.29 | 0.437 |
| | Absent | 5.45 ± 0.38 | | 4.71 ± 1.20 | | 2.51 ± 0.28 | | 2.51 ± 0.35 | |
| *kgp-II* | Present | 5.35 ± 0.52 | <0.0001 * | 4.74 ± 1.49 | 0.004 * | 2.53 ± 0.38 | 0.5267 | 2.50 ± 0.52 | 0.3906 |
| | Absent | 5.8 ± 0.56 | | 5.37 ± 1.16 | | 2.50 ± 0.20 | | 2.54 ± 0.24 | |
| *rgpA*, A | Present | 5.70 ± 0.64 | 0.6495 | 5.35 ± 1.21 | 0.04 * | 2.54 ± 0.21 | 0.02 * | 2.56 ± 0.26 | 0.1133 |
| | Absent | 5.65 ± 0.67 | | 4.96 ± 1.58 | | 2.46 ± 0.30 | | 2.48 ± 0.40 | |
| *rgpA*, B | Present | 5.85 ± 1.63 | 0.1621 | 5.48 ± 2.60 | 0.2767 | 2.47 ± 0.55 | 0.3816 | 2.45 ± 0.80 | 0.1526 |
| | Absent | 5.66 ± 0.48 | | 5.16 ± 1.05 | | 2.52 ± 0.19 | | 2.54 ± 0.23 | |
| *rgpA*, C | Present | 5.40 ± 1.41 | 0.2673 | 4.67 ± 4.80 | 0.3114 | 2.30 ± 0.40 | 0.02 * | 2.35 ± 1.44 | 0.1361 |
| | Absent | 5.69 ± 0.48 | | 5.22 ± 0.99 | | 2.52 ± 0.18 | | 2.54 ± 0.22 | |

PD: probing depth, CAL: clinical attachment loss, PI: plaque index, GI: gingival index, SD: standard deviation, unpaired *t*-test. * *p* < 0.05.

**Table 4.** Association of genotypes of gingipain with the quantity of *P. gingivalis* in chronic periodontitis patients.

| Gingipain Genotypes | Presence/ Absence | Mean | ±SEM | Median | Interquartile Range (IQR) | *p* Value |
|---|---|---|---|---|---|---|
| *kgp-I* | Presence | $2.11 \times 10^8$ | $1.02 \times 10^8$ | $3.85 \times 10^6$ | $2.26 \times 10^7$ | <0.0001 * |
| | Absence | $7.51 \times 10^7$ | $7.19 \times 10^7$ | $1.89 \times 10^4$ | $5.24 \times 10^6$ | |
| *Kgp-II* | Presence | $1.36 \times 10^8$ | $1.30 \times 10^8$ | $4.53 \times 10^6$ | $1.01 \times 10^7$ | 0.1378 |
| | Absence | $1.51 \times 10^8$ | $7.41 \times 10^7$ | $1.17 \times 10^6$ | $1.09 \times 10^7$ | |
| *rgpA*, type A | Presence | $1.57 \times 10^8$ | $7.21 \times 10^7$ | $4.07 \times 10^6$ | $6.81 \times 10^7$ | <0.0001 * |
| | Absence | $1.32 \times 10^8$ | $1.23 \times 10^8$ | 0.0000 | $1.23 \times 10^8$ | |
| *rgpA*, type B | Presence | $3.68 \times 10^8$ | $3.44 \times 10^8$ | $4.13 \times 10^6$ | $3.40 \times 10^8$ | 0.0715 |
| | Absence | $1.13 \times 10^8$ | $5.24 \times 10^7$ | $1.35 \times 10^6$ | $5.10 \times 10^7$ | |
| *rgpA*, type C | Presence | $9.39 \times 10^6$ | $2.79 \times 10^6$ | $1.19 \times 10^7$ | $9.06 \times 10^6$ | 0.1378 |
| | Absence | $1.52 \times 10^8$ | $6.63 \times 10^7$ | $1.59 \times 10^6$ | $6.47 \times 10^7$ | |

SEM: standard error of mean. Mann Whitney U test, * *p* < 0.05.

The quantitative relationshipsfor the level of *P. gingivalis* with the gingipain genotypes is depicted in Figure 2. A detection frequency of >$10^7$ cells of *P. gingivalis* was higher in the presence of *kgp-I* (35.94%) when compared to the presence of *kgp-II* (25.81%). The detection frequency of ≤$10^4$ cells of *P. gingivalis* was higher in the presence of *kgp-II* (25.81%) when compared with any other genotype.

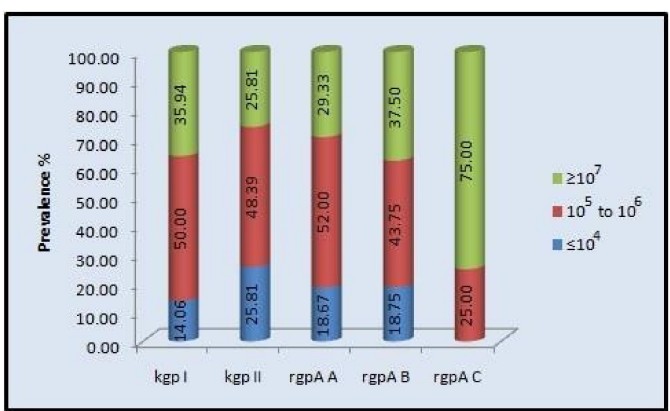

**Figure 2.** Relationship between *P. gingivalis* level and prevalence of *kgp* and *rgpA* genotypes in chronic periodontitis patients.

## 4. Discussion

Gingipain-containing *P. gingivalis* have been linked to systemic chronic inflammatory diseases and have been associated with periodontal disease development [17]. Thus, virulent factors are important determinants of the pathogenic ability of *P. gingivalis*. To investigate the pathogenic variability of *P. gingivalis*, it is necessary to study the association between these virulence factors and diseased and healthy conditions. Fimbriae genotyping has been performed extensively across different categories of population and the association of specific *fimA* virulent types with disease and health is well-documented [18–20].

We found that the most prevalent *kgp* type in the CP group was type I (67.5%), whereas, in the H group, *kgp-II* was the most frequently found gingipain (97.14%). The genotype *kgp-II* was significantly associated with health condition. Among *rgpA* genotypes, type A was most frequently found in both the groups (78.95% and 82.85%, respectively). Beikler et al. also reported a comparable frequency of lysine-specific protease (kgp) genotypes in a German population [8]. They found that *kgp-I* was colonized in 57.1% of the *P. gingivalis* positive patients and *kgp-II* in 42.2% of the positive patients. Allaker and his co-workers found *rgpA* type A was the most prevalent (77%) arginine-specific gingipain in chronic periodontitis patients, followed by type B and type C.They also found both type A and B at two different sites from a single individual [7]. Our findings did not indicate the presence of any such combinations. Another study by Yoshino et al. found that *kgp* type I and type II occurred at a similar frequency (56.5% and 43.5%, respectively) in isolates of chronic periodontitis patients. Among *rgpA* genotypes, type A was found in 75.8% of the 62 isolates, followed by type B and type C, which were detected in 21% and 3% of the clinical isolates [21]. A study by Abusleme L found that *kgp-I* was prevalent in 65.2% of the samples and *kgp-II* in 34.8% of the samples. All the 38 isolates had *rgpA* type A (100%), with no detection of type B and type C [22]. All these studies were performed on either chronic periodontitis patients or in aggressive periodontitis cases. Due to the dominant occurrence of *rgpA*, type A in both the groups, this study could not establish an association of any specific arginine-specific gingipain with diseased or health condition. Significant results were obtained regarding lysine-specific gingipain, with *kgp-I* and *kgp-II* being frequently associated with diseased and health condition, respectively.

The relationship of gingipain genotypes with clinical parameters showed that the presence of *kgp-I* was frequently associated with deep pocket depth and more clinical attachment loss compared to other types. Furthermore, *kgp-II* was more frequently detected in shallow pocket depths and, comparatively, there was less clinical attachment loss when compared to other types. In contrast to our results, Beikler et al. did not find any association of gingipains with clinical characteristics, such as probing depth and bleeding on probing [8]. Nevertheless, we agree that the higher prevalence of *kgp-I* observed in the CP group suggests its better adaptability to environmental changes and, hence, its capacity to enable *P. gingivalis* colonization of the oral cavity.

Oral biofilm formation is the initial step in the progression of chronic periodontitis disease. Streptococci species, the known early colonizers in plaque formation, are major pathogens that colonize the adherent rich plaque. The ability of *P. gingivalis* to adhere to *Streptococcus gordonii* under the stress of low shear forces, and to form biofilm structures typical of biofilm deposits, has been investigated in an invitro model by Cook GS et al. [23]. They found that *P. gingivalis* exhibited preferential adherence to a streptococcal surface compared to a saliva-coated glass surface. In another study, it was found that *Streptococci* played a vital role in facilitating the establishment of *P. gingivalis* by depletion of environmental oxidants [24]. In our study, the *rgpA*, type A genotype was found to be significantly associated with increased plaque index (PI) and increased clinical attachment loss (CAL). This correlation was strengthened by the increased colonization of *P. gingivalis* in the CP group in the presence of the same genotype. This leads us to believe that there could be vital interactions between *rgpA*, type A with early colonizers, which can benefit *P. gingivalis* in the colonization process. The relationship of this genotype with different early colonizers needs to be further explored.

Tomoko Kadowaki used electron microscopy to examine the fimbriation process of *P. gingivalis* using KGP- and RGP-deficient mutant strains [25]. They found that an RGP-null mutant strain showed little or no fimbriation whereas a KGP-deficient mutant possessed characteristic fimbriae on the cell surface. This suggests that RGP, and not KGP, is responsible for the fimbriation of *P. gingivalis*. Our results, which showed a significant relationship of *rgpA*, type A with increased colonization of *P. gingivalis*, further establish the role of this gingipain genotype in bacterial adhesion through the processing of fimbrilin. It is possible that colonization of certain pathogenic genotypes could lead to progression of the co-colonization of other putative pathogens and disease progression. The exact mode of interaction among these players needs to be further investigated.

The clinical characteristics correspond to our quantitative data in which the mean number of cells of *P. gingivalis* was increased with the presence of *kgp-I* and *rgpA* type A (*p* value < 0.0001 for both). There was also evidence of *kgp-II* being associated with health condition, as the highest percentage (25.8%) of samples showed <$10^4$ cells of *P. gingivalis* when this genotype (*kgp-II*) occurred in the patient compared to other genotypes. Although there is sequence variation in the two *kgp* types, the differential activity of these two *kgp* types—one playing a role in periodontitis, the other in healthmay be explained by the differing adherence capacity of each genotype mediated by different *kgp* products exhibiting differential substrate specificity.

Gingipains have been implicated in other diseases, which highlights the urgent need for further research in this area. A recent study suggested that *P. gingivalis* that express *Kgp* and *Rgp*, attaches to the tracheal and bronchial pulmonary tissue and induces mucin production leading to aggravation of chronic obstructive pulmonary disease (COPD) [26]. Gingipain virulent factors have been known to manipulate the immune system, causing immuno suppression that triggers neurodegenerative processes related to Alzheimer's disease [27]. Gingipain inhibition may provide multiple benefits for periodontal disease treatment and prevention. In one of our previous studies, we found that resveratrol was able to inhibit gingipain gene expression, thereby limiting the formation of mature *P. gingivalis* biofilm in an invitro study [28]. Other studies have shown that antibodies targeted against the amino terminal region of the catalytic domain of gingipain R influence the immunogenic response against *P. gingivalis*, suggesting the potential availability of vaccination for human use [29]. Studies have found that tetracycline and its analogues improve clinical characteristics in patients with chronic periodontitis by inhibiting gingipain activities [30]. The results obtained from our study suggest that gingipain genotyping can be utilized as an important determinant of periodontal health and disease condition and further establishes its role in the pathophysiology of periodontitis, indicating its utility as a target in periodontal therapy.

One of the limitations of this study is that we employed a pooled sample technique; therefore, the co-existence of different genotypes in the same ecological niche could not be assessed. The individual site needs to be evaluated to relate colonization to a specific ecological niche by a certain genotype.

## 5. Conclusions

A unique finding of this study is the positive association of periodontal health with the *kgp-II* genotype. The significant association of the *kgp-I* genotype with chronic periodontitis confirms findings from previously reported studies. We also found no polymorphism existed for the *rgpA* gene, but that *rgpA*, type A played an important role in bacterial colonization. However, the exact mode of *rgpA*, type A interaction with other colonizing bacterial needs to be investigated. In future, larger randomized control trials are needed to confirm the findings of this study. This will help in the quest to find possible markers for periodontal health and disease.

**Author Contributions:** Conceptualization, M.K., K.B. and U.M.; methodology, M.K.; writing— original draft preparation, M.K. and V.J.; writing—review and editing, M.R.P. and V.K. All authors have read and agreed to the published version of the manuscript.

**Funding:** This research received no external funding.

**Institutional Review Board Statement:** The study was approved by the Institutional Review Board of Maratha Mandal's NGH Institute of Dental Sciences and Research Centre, Belagavi (Certificate no. 2016/819).

**Informed Consent Statement:** Informed consent was obtained from all subjects involved in the study.

**Data Availability Statement:** The data presented in this study are available on request from the corresponding author.

**Acknowledgments:** The authors would like to thank Basavaraj Hungund, Department of Biotechnology, KLE Technological University for supporting this study. We would also like to thank Ramakant Nayak, Principal, Maratha Mandal's NGH Institute of Dental Sciences and Research Centre, Belagavi for allowing us to conduct this study.

**Conflicts of Interest:** The authors declare no conflict of interest.

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
