# Peer review of "Gingipain Genotyping as a Potential Predictor for the Assessment of Periodontal Health and Disease Condition"

_2673-947X, doi:10.3390/hygiene2040016_

Round 1

Reviewer 1 Report

Wht does this article bring new from thr other one already published?

In the discussion chapter i suggest the authors to incvlude more recent published articles and compare their results with the ones from other articles published in the literature.

Please also mention the limitations of your study.

Author Response

We thank all the reviewers for their constructive comments to improve the quality of this research paper. We have made necessary changes as per the suggestions. The changes have been marked in the manuscript file. Below are the point wise clarifications to the changes made to the original manuscript.

Thank you

Reviewer 1:

Wht does this article bring new from the other one already published?

Ans: The study was aimed at exploring gingipain genotypes in chronic periodontitis and healthy individuals. Although there are numerous studies on distribution of gingipain in diseased cases, there are very few studies in which the role of gingipains in healthy individuals was explored. With this objective in mind, we could find that kgp-I was associated with diseased condition whereas kgp-II was associated with health. Moreover rgpA, type A plays a role in the colonization of P. gingivalis. These are some of the novel findings which adds to the already available information.

In the discussion chapter i suggest the authors to include more recent published articles and compare their results with the ones from other articles published in the literature.

Ans: As per the suggestions we have made some additions to the discussion section with recent published articles (New reference numbers 17, 26, 27, 28)

Please also mention the limitations of your study.

Ans: One of the limitations of this study is that we employed pooled sample technique therefore co-existence of different genotypes in the same ecological niche could not be analyzed. The individual site needs to be evaluated in order to relate colonization to a specific ecological niche by a certain genotype.

Now this has been added to the discussion section last paragraph.

Reviewer 2 Report

Dear authors,

This study was designed to determine if the gingipain genotyping could be a potential predictor for the assessment of periodontal health and disease condition. For this, they used PCR and focuss on 4 genotypes. Their results demonstrated that kgp-I and kgp-II were strongly associated with disease and health condition respectively. 

Major comment

Even if it is not the first publication with this database. It could be interesting to report data according to CONSORT and to add the checklist in supplementary file. 

Minor comment

Line 17 “individualspos-“ Add one space after “individuals

Line 125 Replace « reference » by the right number.

Author Response

We thank all the reviewers for their constructive comments to improve the quality of this research paper. We have made necessary changes as per the suggestions. The changes have been marked in the manuscript file. Below are the point wise clarifications to the changes made to the original manuscript.

Thank you

Major comment

Even if it is not the first publication with this database. It could be interesting to report data according to CONSORT and to add the checklist in supplementary file. 

Ans: This is not a randomized clinical control study; hence we feel that reporting the data according to CONSORT would not be right.

Minor comment

Line 17 “individualspos-“ Add one space after “individuals

Ans: Space is added.

Line 125 Replace « reference » by the right number.

Ans: Reference no. 16 is added.

Reviewer 3 Report

The authors aimed to investigate the occurrence of gingipain genotypes of P. gingivalis in patients with chronic periodontitis and healthy individuals and assess their association with clinical characteristics in these subjects.

The study covers some issues that have been overlooked in other similar topics. The structure of the manuscript appears adequate and well divided in the sections. Moreover, the study is easy to follow, but some issues should be improved. Some of the comments that would improve the overall quality of the study are:

a. Authors must pay attention to the technical terms acronyms they used in the text.

b. English language needs to be revised.

c. Limitations of the study needs to be added.

d. Conclusion Section: This paragraph required a general revision to eliminate redundant sentences and to add some "take-home message".

Author Response

We thank all the reviewers for their constructive comments to improve the quality of this research paper. We have made necessary changes as per the suggestions. The changes have been marked in the manuscript file. Below are the point wise clarifications to the changes made to the original manuscript.

Thank you

  1. Authors must pay attention to the technical terms acronyms they used in the text.

Ans: Changes have been made.

  1. English language needs to be revised.

Ans:  English language has been revised.

  1. Limitations of the study needs to be added.

Ans: One of the limitations of this study is that we employed pooled sample technique therefore co-existence of different genotypes in the same ecological niche could not be analyzed. The individual site needs to be evaluated in order to relate colonization to a specific ecological niche by a certain genotype.

Now this has been added to the discussion section last paragraph.

  1. Conclusion Section: This paragraph required a general revision to eliminate redundant sentences and to add some "take-home message".

Ans: The conclusion section has been revised.  

The unique finding of this study has been the positive association of periodontal health to kgp-II genotype. The significant association of kgp-I genotype with chronic periodontitis confirms the findings from earlier reported studies. We also found no polymorphism existed for rgpA gene, but rgpA, type A played aimportant role in bacterial colonization. However, the exact mode of rgpA, type A interactions with other colonizing bacterial needs to be investigated. In future larger randomized control trials are needed to confirm the findings of this study. This certainly will help in quest to find possible markers for periodontal health and disease.